# Can forensic radiological skeletal age estimation be performed by examining ischiopubic-ilioischial-iliopubic synchondrosis in computed tomography images?

**Burak Gümüş**[1]*, **Erdal Karavaş**[2], **Onur Taydaş**[3]

**1** Faculty of Medicine, Department of Forensic Medicine, Hitit University, Çorum, Turkey, **2** Faculty of Medicine, Department of Radiology, Binali Yildirim Erzincan University, Erzincan, Turkey, **3** Faculty of Medicine, Department of Radiology, Sakarya University, Adapazarı, Turkey

* eti19030@hotmail.com

## Abstract

### Introduction

In this study, we evaluated whether it is possible to perform forensic radiological skeletal age estimation via radiological examination of the ilioischial, ischiopubic, and iliopubic synchondrosis regions of the pelvis.

### Methods

This study was conducted by retrospectively examining the abdominopelvic images of individuals aged 8–16 who had applied to the hospital for any reason without having a chronic disorder and who had undergone computed tomography. Two radiologists retrospectively reviewed the images. The BT images of the pelvis ilioischial, ischiopubic, and iliopubic synchondrosis regions were evaluated as follows: 0: open, 1: semiclosed, and 2: closed. The data were evaluated using the SPSS 17 program.

### Results

Two hundred sixty-three children (118 girls and 145 boys) between the ages of 8 and 16 years without any health problems participated. There was a significant difference between the groups for all the evaluated synchondrosis joints in girls and boys (p<0.001 for each group comparison). We observed that ilioischial, ischiopubic, and iliopubic synchondrosis closed earlier in girls than boys. In addition, we found that the joints were closed at the age of 15 and over in boys and at 14 and over in girls.

### Discussion

Some studies have previously evaluated synchondrosis by using computed tomography. We showed that forensic radiological skeletal age estimation could be performed by examining ischiopubic-ilioischial-iliopubic synchondrosis in pelvis computed tomography images. The pelvis is more resistant to decay than other parts of the body. Furthermore, pelvis

**Data Availability Statement:** All relevant data are within the paper and its Supporting Information files.

**Funding:** The authors received no specific funding for this work.

**Competing interests:** The authors have declared that no competing interests exist.

bones can withst and the effects of postmortem animal attacks for a longer period. There-fore, we believe that forensic age estimation can be made on corpses with no extremity, a damaged chest, or whose only pelvic bones are assessable through the method we use.

## Conclusion

In our study, the ischiopubic-ilioischial-iliopubic joints were open in those aged nine and under and closed in those aged 15 and above. Ilioischial, ischiopubic, and iliopubic synchon-drosis were observed to close earlier in girls than in boys. We consider that our study will be beneficial in the 8-16-year-old age group if used. In addition, our study can be used to deter-mine the radiological bone age in cases with wrist bone abnormalities or wrist amputation.

## Introduction

For age estimation, topics related to physical development or aging have been researched. The criteria used for age estimation include height, weight, development of secondary sex charac-teristics, skin and eye changes, hair, tooth and bone development, and psychological develop-ment [1,2]. Bone age is an indicator of skeletal maturity. [3]. Comparison of bone age with chronological age; It is important in pediatric endocrinology, in the diagnosis of orthodontic and orthopedic diseases, or in societies where there is a lack of documents and birth records are not kept properly [4–6]. Because immigrants' birth records and population information may not be available due to the increasing number of worldwide armed conflicts, there is a growing interest in correct age estimation techniques in many countries. Additionally, age esti-mation is requested by the courts and governmental institutions in cases where the individual's age is unknown and legally suspected. Age estimation of an individual might be required in cases such as criminal liability in forensic practices, legal capacity, and the ability to perceive the legal meaning and consequences of an act and direct their behavior, recruitment, entering civil service, marriage, and retirement [1,7].

Bone age is a valuable indicator that shows the maturity of an individual's skeleton system, identifies and diagnoses various diseases, and determines the timing of treatment. Therefore, the accuracy of the bone age evaluation is crucial. Estimating bone age becomes more an issue, especially in the pubertal period [2,8,9].

Many rights and responsibilities for children are associated with adopting the Convention on the Rights of the Child and legal age limits. For example, governments may allow children to marry, consent to sexual relations, and accept or refuse health services before reaching adulthood. The age at which criminal responsibility is acquired may precede the legal age-based definition of adulthood and range from 7 to 16 years [10].

Bone age assessment is usually compared with standard radiological left wrist bone develop-ment atlases [11]. The formation and development of growth plates in the bones, the radiologi-cal examination of the bones based on the method of finding the pineal and diaphysis lines and ossification points, and their adaptation to the existing atlases remain important as the method that is used in the clinic and closest to the actual values [12]. In addition, there are researches on the determination of bone age using imaging methods such as computed tomog-raphy, magnetic resonance imaging, and ultrasound [13].

This study aimed to demonstrate that forensic radiological skeletal age estimation can be performed by examining the synchondrosis regions of the ilioischial, ischiopubic, and iliopu-bic bones in the pelvis.

## Methods

This study is conducted by retrospectively examining the abdominopelvic images of individuals chronologically aged from 8–16 who have applied to the hospital for any reason without having any chronic disorder and have undergone computed tomography (Our study was carried out using data between the years 2016–2021). Two hundred sixty-three children (118 girls and 145 boys) between the ages of 8 and 16 years who had no health problems participated in the study. According to the Helsinki Charter, the Ethics Committee of Erzincan Binali Yıldırım University, Turkey, approved this study, and it was conducted in the radiology and forensic medicine department of the medical faculty.

As physiological growth and development processes change according to sex, all calculations were performed separately for girls and boys. We performed age grouping at the following intervals: we formed seven different age subgroups between 8 and 16 years.

Abdominopelvic computed tomography (CT) scans were performed with a 16-slice CT scanner (Sensation 16, Siemens Medical Systems, Forchheim, Germany). No oral contrast material was administered. Abdominopelvic CT scans (tube voltage = 120 kV, effective mAs = 90, slice thickness 5 mm, collimation = 2x4 mm, pitch = 1.6, and imaging reconstruction performed in the axial sagittal plane with a slice thickness of 1.5 mm) were acquired without an intravenous (IV) or oral contrast. The body region between the level of the upper diaphragm and the greater trochanter was included in the abdominopelvic CT scan area. Images were reviewed on the Syngo.via software (Siemens Healthcare, Forchheim, Germany) before and after the creation of virtual rendering technique (VRT) images (Figs 1 and 2). Two radiologists retrospectively reviewed the images, one with ten years of experience and the other with five years of experience. There was an agreement between them for each case.

In the BT, images of the pelvis ilioischial, ischiopubic, and iliopubic synchondrosis regions were evaluated as follows: 0: open, 1: semiclosed, and 2: closed.

The data were evaluated using the SPSS 17 program. Bar graphs illustrate the frequency distributions in groups. The groups were independent, and the variables were categorical. Thus, group comparisons were made using the chi-squared test. The confidence intervals were set to 95%, and probability values (p) <0.05 were considered statistically significant. This study's power was calculated using the PASS 12 program (NCSS, UT, USA).

All data in the study were analyzed anonymously and presented anonymously. No personal data of any individual is stored or presented. Our study is a retrospective and observational study, and there is no ethical requirement to obtain consent from the participants or parents.

## Results

The radiological screening results obtained from each evaluated joint are presented in Tables 1 and 2 separately for girls and boys. The bar graphs illustrate the distribution of the synchondrosis joint spacing or closure status of each scanned joint by group (Figs 7 and 8). Since the number of data points in the cells remained below 5, whether there was a difference between the groups was evaluated according to Fisher's exact test. It was observed that there was a significant difference at the p <0.001 level between all groups in both girls and boys (Tables 1 and 2).

This study's power with the current design, which contains seven groups, was calculated to be 0.95 using the chi-squared test procedure for all pairs. The type I error (α value or p-value) was set at 0.05. The power of the present study is computed from the results after the study is completed. The study, which was classified as a retrospective study, included children who had computed lower abdomen and pelvis computed tomography for any reason and had no

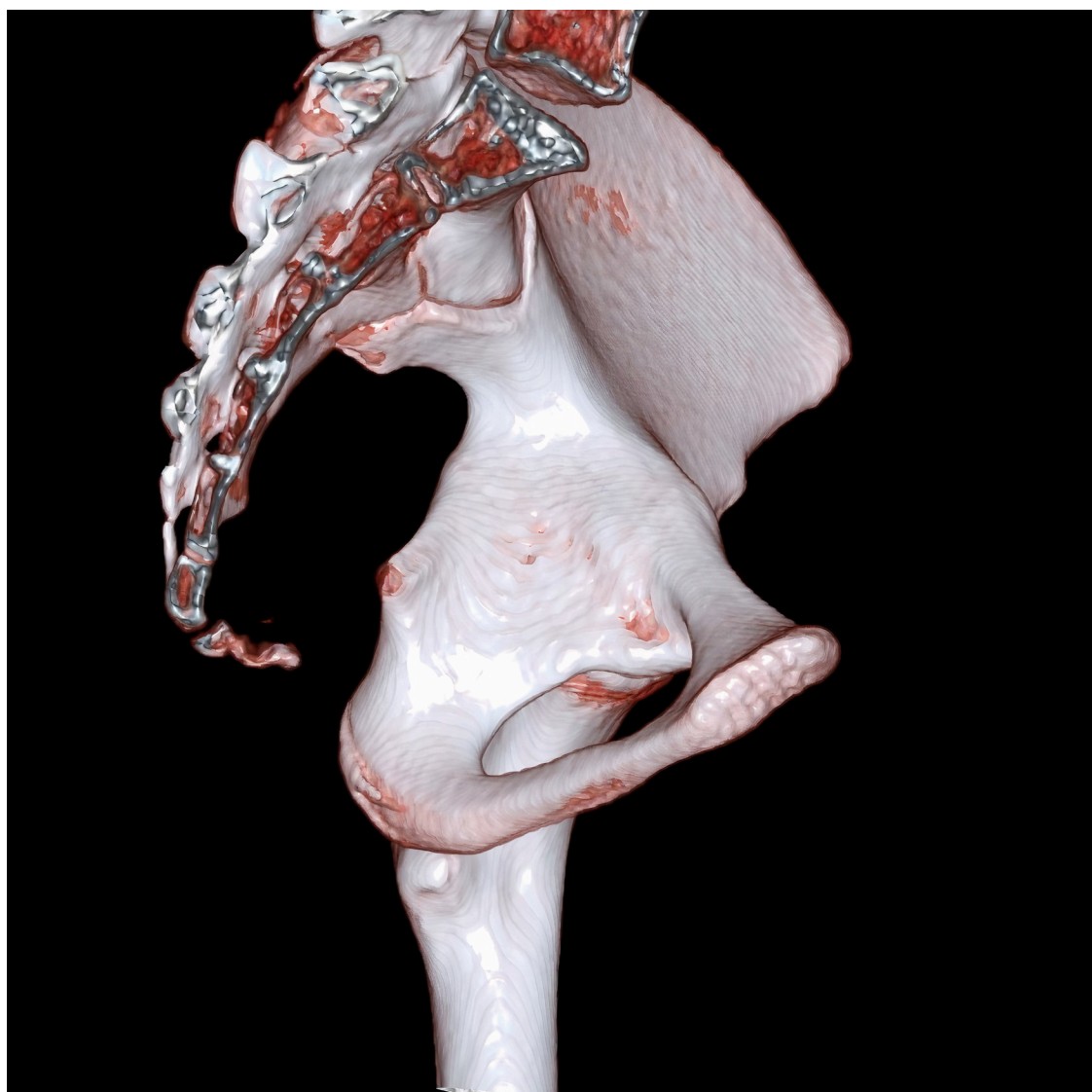

**Fig 1. For a patient who is a 16-year-old girl, the virtual rendering technique (VRT) computed tomography image shows that all three synchondrosis regions are closed.**

chronic disease. There are 263 children in our study. A total of 118 (44.9%) were girls, and 145 (55.1%) were boys.

There must be a regular change in the figure from opening to closing. The Fig 7 shows that girls' ischiopubic, ilioischial, and iliopubic synchondrosis regions are closed at 13. Girls' ischiopubic, ilioischial, and iliopubic synchondrosis regions are semiclosed at 10–12 years. Besides, in girls, ilioischial-iliopubic-ischiopubic synchondrosis regions opened at the age of 8–9 years.

There must be a regular change in the figure from opening to closing. The Fig 8 shows that the ilioischial-iliopubic-ischiopubic synchondrosis regions in boys are closed at the age of 15–16. The ischiopubic, ilioischial, iliopubic, and synchondrosis regions in boys are semiclosed at the ages of 11–14 years. In addition, 8-10-year-old boys did not have any closure in their ilioischial-iliopubic-ischiopubic synchondrosis regions.

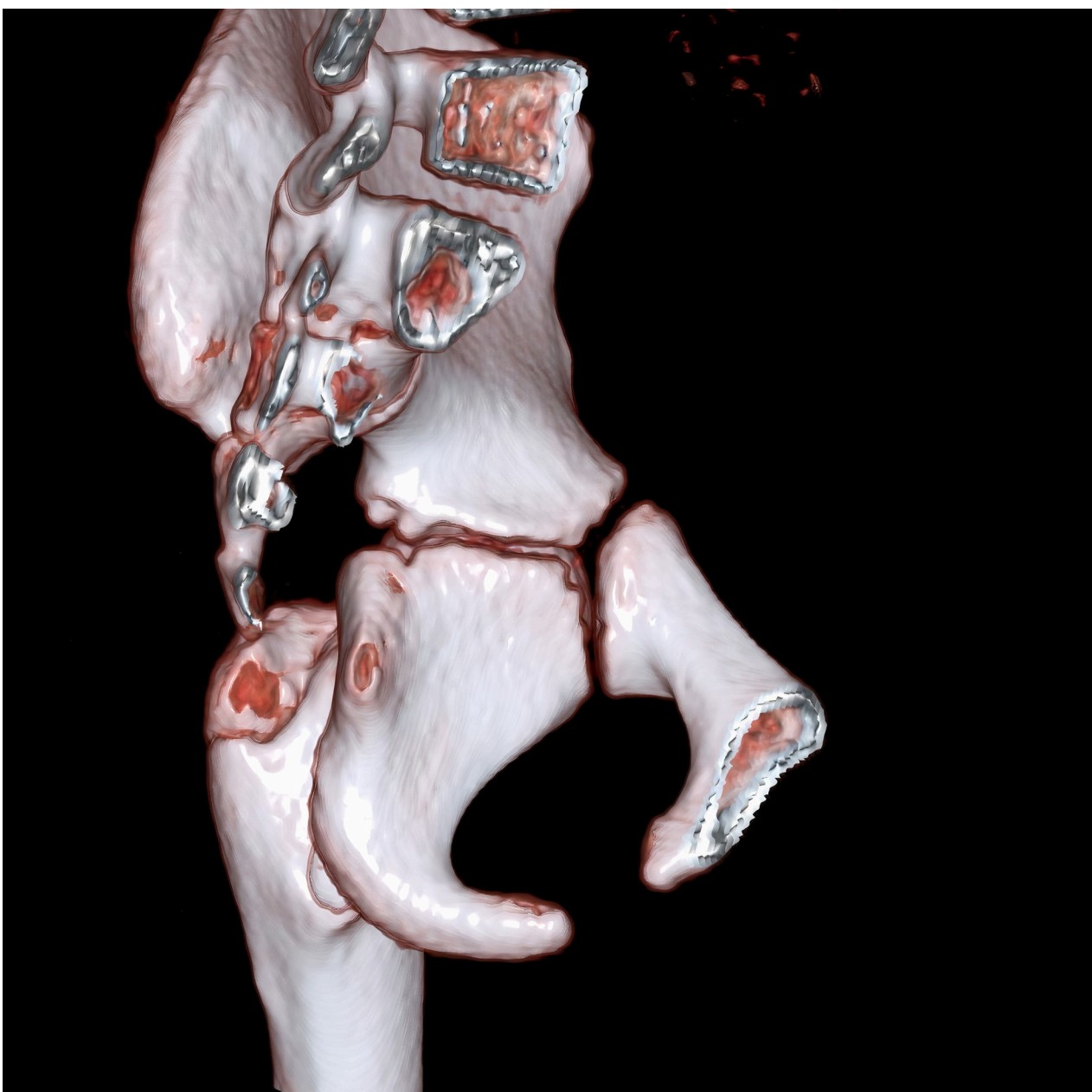

**Fig 2. For a patient who is an 8-year-old girl, the virtual rendering technique (VRT) computed tomography image shows that all three synchondrosis regions were open.**

The table shows that the joints of the iliopubic synchondrosis junction point are generally open in girls who are 8–9 years old and all closed at the ages of 14–16 years old. The joints of the ischiopubic-ilioischial synchondrosis junction point are generally open in children who are 8-9-year-old girls. The table shows that girls over 13 years of age do not have a fully open junction in the ischiopubic-ilioischial-iliopubic synchondrosis regions. The table also shows that there were statistically significant differences between the groups.

The table shows that ischiopubic-ilioischial-iliopubic synchondrosis junction points were open in all boys aged 8–9 years. In addition, ischiopubic-iliopubic synchondrosis junction

**Table 1. Ischiopubic, ilioischial, and iliopubic synchondrosis junction points in girls by age.**

| Girls | Ischiopubic | | | | | | Ilioischial | | | | | | Iliopubic | | | | | |
|---|---|---|---|---|---|---|---|---|---|---|---|---|---|---|---|---|---|---|
| | 0 | | 1 | | 2 | | 0 | | 1 | | 2 | | 0 | | 1 | | 2 | |
| | N | % | N | % | N | % | N | % | N | % | N | % | N | % | N | % | N | % |
| 8–9 | 11 | 68.8 | 3 | 18.8 | 2 | 12.4 | 15 | 94.8 | 1 | 6.2 | - | - | 16 | 100 | - | - | - | - |
| 10 | 11 | 73.3 | 4 | 26.7 | - | - | 9 | 60 | 5 | 33.3 | 1 | 6.7 | 14 | 93.3 | 1 | 6.7 | - | - |
| 11 | 5 | 31.3 | 9 | 56.3 | 2 | 12.4 | - | - | 6 | 37.5 | 10 | 62.7 | 5 | 31.3 | 8 | 50 | 3 | 18.8 |
| 12 | - | - | 10 | 67.7 | 5 | 33.3 | - | - | 1 | 6.7 | 14 | 93.3 | - | - | 8 | 53.3 | 7 | 46.7 |
| 13 | - | - | 1 | 6.7 | 14 | 93.3 | - | - | - | - | 15 | 100 | - | - | 1 | 6.7 | 14 | 93.3 |
| 14 | - | - | - | - | 17 | 100 | - | - | - | - | 17 | 100 | - | - | - | - | 17 | 100 |
| 15–16 | - | - | - | - | 26 | 100 | - | - | - | - | 26 | 100 | - | - | - | - | 26 | 100 |
| Group Comparison | p< 0.001 | | | | | | p< 0.001 | | | | | | p< 0.001 | | | | | |

0: Joint open, 1: Joint closed, and 2: Joint semiclosed. Group comparisons were made according to Fisher's Exact Test with the Chi-Squared test for the 7x3 table (as the smallest expected value for all three parameters was below 5. (the significance values = 0.000)).

points are open, and the iliopubic synchondrosis junction point is generally open in all boys aged ten years. The table also shows that boys over 14 years of age do not have fully open junctions in ischiopubic-ilioischial-iliopubic synchondrosis regions. Furthermore, in all the children aged from 15–16, there were closed ischiopubic-ilioischial-iliopubic synchondrosis joints; there were no closed synchondrosis joints at the ischiopubic and iliopubic junctions for children age 11 and below, and there was no closed synchondrosis under the age of 10 for ilioischial regions. The table shows that there were statistically significant differences between the groups.

## Discussion

Physical, sexual, skeletal, and dental maturity indicators are frequently used for age estimation [14]. Hand and wrist radiography is the most commonly used method to evaluate skeletal maturity or bone age [15,16]. For many years, the Radiographic Atlas of Skeletal Development of the Hand and Wrist by Greulich and Pyle or Tanner-Whitehouse Atlas has been used for age estimation depending on the bones' presence, size, and shape in hand and wrist [17,18]. This study presented the subject by using pelvis computed tomography (Figs 1–6) for forensic

**Table 2. Ischiopubic, ilioischial, and iliopubic synchondrosis junction points in boys by age.**

| Boys | Ischiopubic | | | | | | Ilioischial | | | | | | Iliopubic | | | | | |
|---|---|---|---|---|---|---|---|---|---|---|---|---|---|---|---|---|---|---|
| | 0 | | 1 | | 2 | | 0 | | 1 | | 2 | | 0 | | 1 | | 2 | |
| | N | % | N | % | N | % | N | % | N | % | N | % | N | % | N | % | n | % |
| 8–9 | 16 | 100 | - | - | - | - | 16 | 100 | - | - | - | - | 16 | 100 | - | - | - | - |
| 10 | 15 | 100 | - | - | - | - | 11 | 73.3 | 4 | 26.7 | - | - | 15 | 100 | - | - | - | - |
| 11 | 17 | 85 | 3 | 15 | - | - | 10 | 50 | 6 | 30 | 4 | 20 | 17 | 85 | 3 | 15 | - | - |
| 12 | 10 | 52.6 | 8 | 42.1 | 1 | 5.3 | 4 | 21.1 | 11 | 57.9 | 4 | 21.1 | 12 | 63.2 | 5 | 26.3 | 2 | 10.5 |
| 13 | 1 | 5 | 17 | 85 | 2 | 10 | 1 | 5 | 7 | 35 | 12 | 60 | 1 | 5 | 10 | 50 | 9 | 45 |
| 14 | - | - | 17 | 73.9 | 6 | 24.1 | - | - | 5 | 21.7 | 18 | 78.3 | 7 | 30.4 | 16 | 69.6 | - | - |
| 15–16 | - | - | 2 | 6.3 | 30 | 93.8 | - | - | 1 | 31.1 | 31 | 96.6 | - | - | 1 | 3.1 | 31 | 96.9 |
| Group Comparison | p< 0.001 | | | | | | p< 0.001 | | | | | | p< 0.001 | | | | | |

0: Joint open: 1: Joint closed, and 2: Joint semiclosed. Group comparisons were made according to Fisher's Exact Test with the Chi-Squared test for the 7x3 table (as the smallest expected value for all three parameters was below 5. (the significance values = 0.000)).

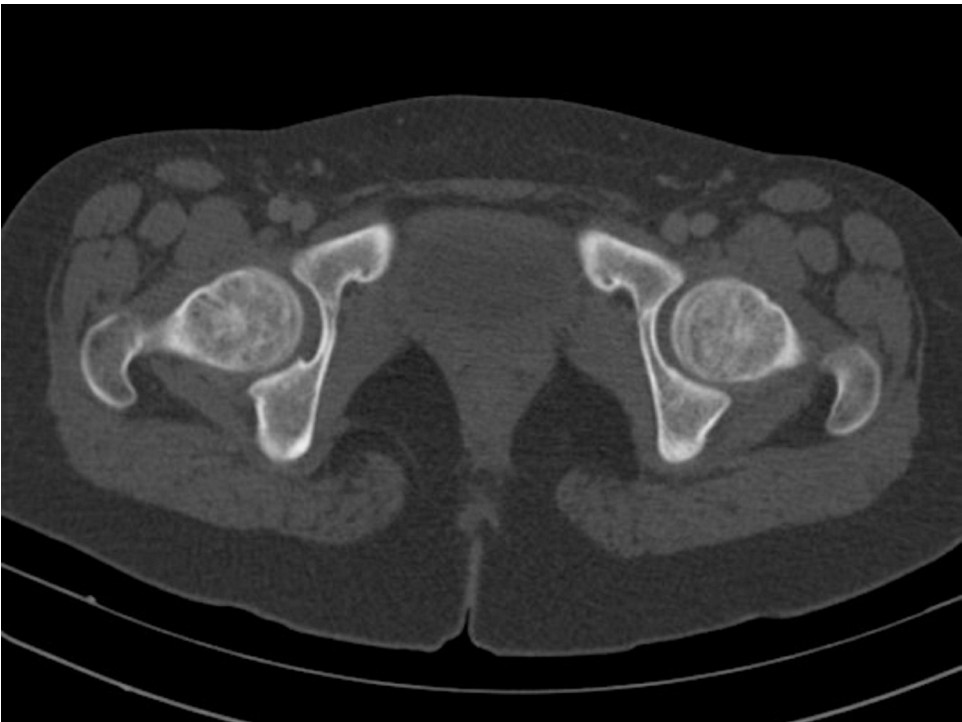

**Fig 3. For a patient who is a 16-year-old boy, the axial plane reformatted computed tomography image shows that all three synchondrosis regions are closed.**

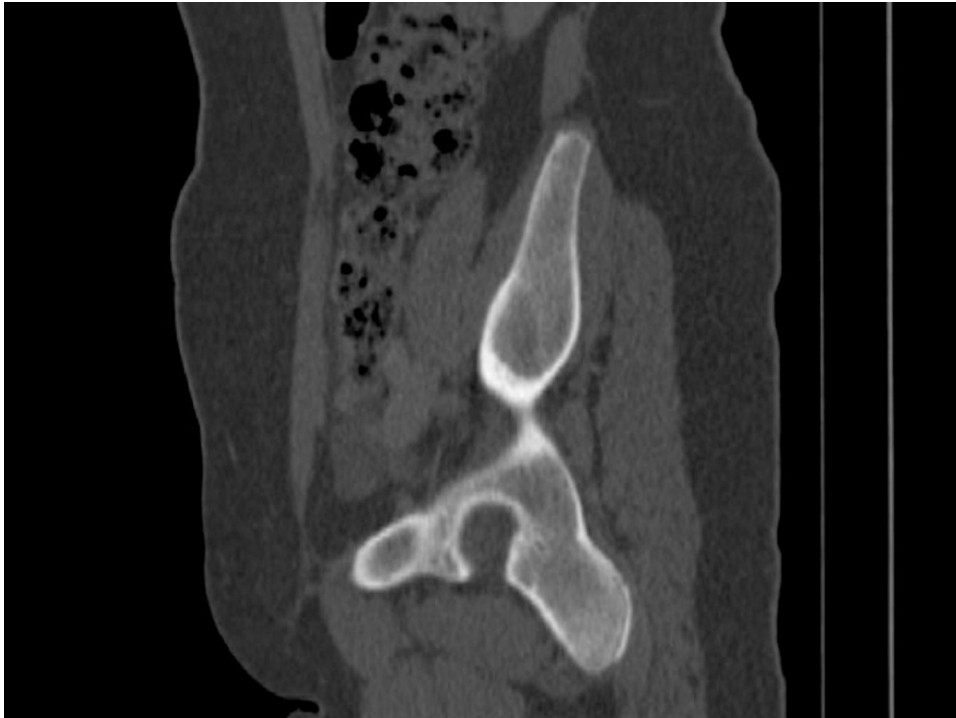

**Fig 4. For a patient who is a 16-year-old boy, the sagittal plane reformatted computed tomography image shows that all three synchondrosis regions are closed.**

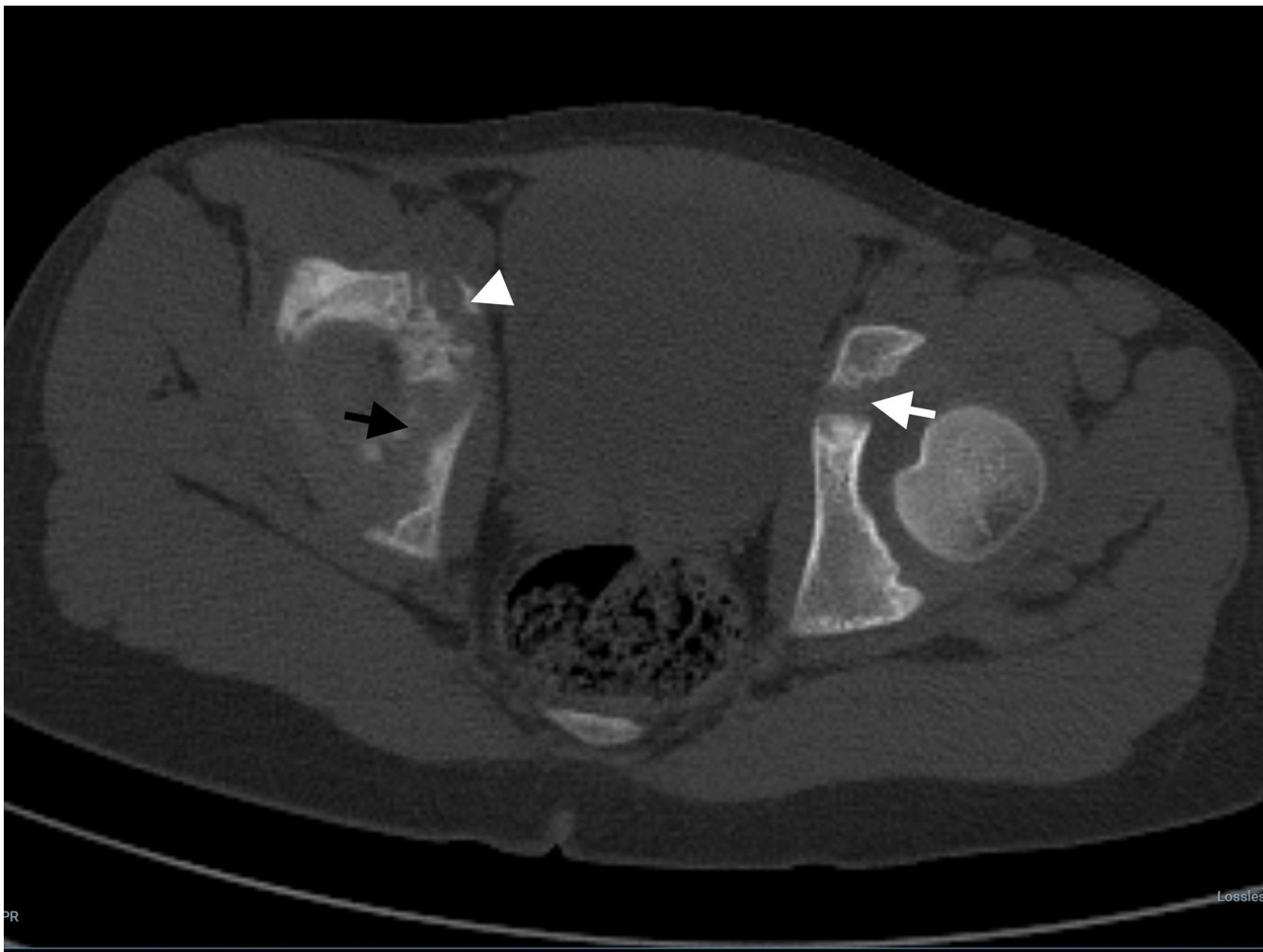

**Fig 5. For a patient who is an 8-year-old girl, the axial plane reformatted computed tomography image shows that all three (white arrow: Ischiopubic, black arrow ilioischial, white arrowhead: Iliopubic) synchondrosis regions are open.**

radiological age estimation. In our study, individuals between 8–16 years of age were involved. The reason for this was that we hypothesized that ischiopubic-ilioischial-iliopubic synchondrosis could be used as a complementary method for forensic radiological age estimation in the 8-16-year-old age group. In order to maintain concordance with previous studies and to obtain more certain results, we classified ischiopubic-ilioiscial-iliopubic synchondrosis as open, semiclosed, and closed [19]. Up to the age of 18, radiological bone age determination is usually made by examining wrist radiography [17]. It can be used in the forensic radiological bone age determination of ischiopubic-ilioischial-iliopubic synchondrosis in people with wrist amputation or wrist development disorders. According to the data we obtained from our study, it is possible to conclude that an individual with closed ischiopubic-ilioischial-iliopubic synchondrosis is 15 years of age and over. Our study shows that boys and girls with open ischiopubic-ilioischial-iliopubic parts were nine years old and below (Tables 1 and 2).

Many studies in the literature show that age determination can be made with computed tomography. Evaluating spheno-occipital synchondrosis using computed tomography has been reported to indicate that significant results were obtained in terms of age estimation.

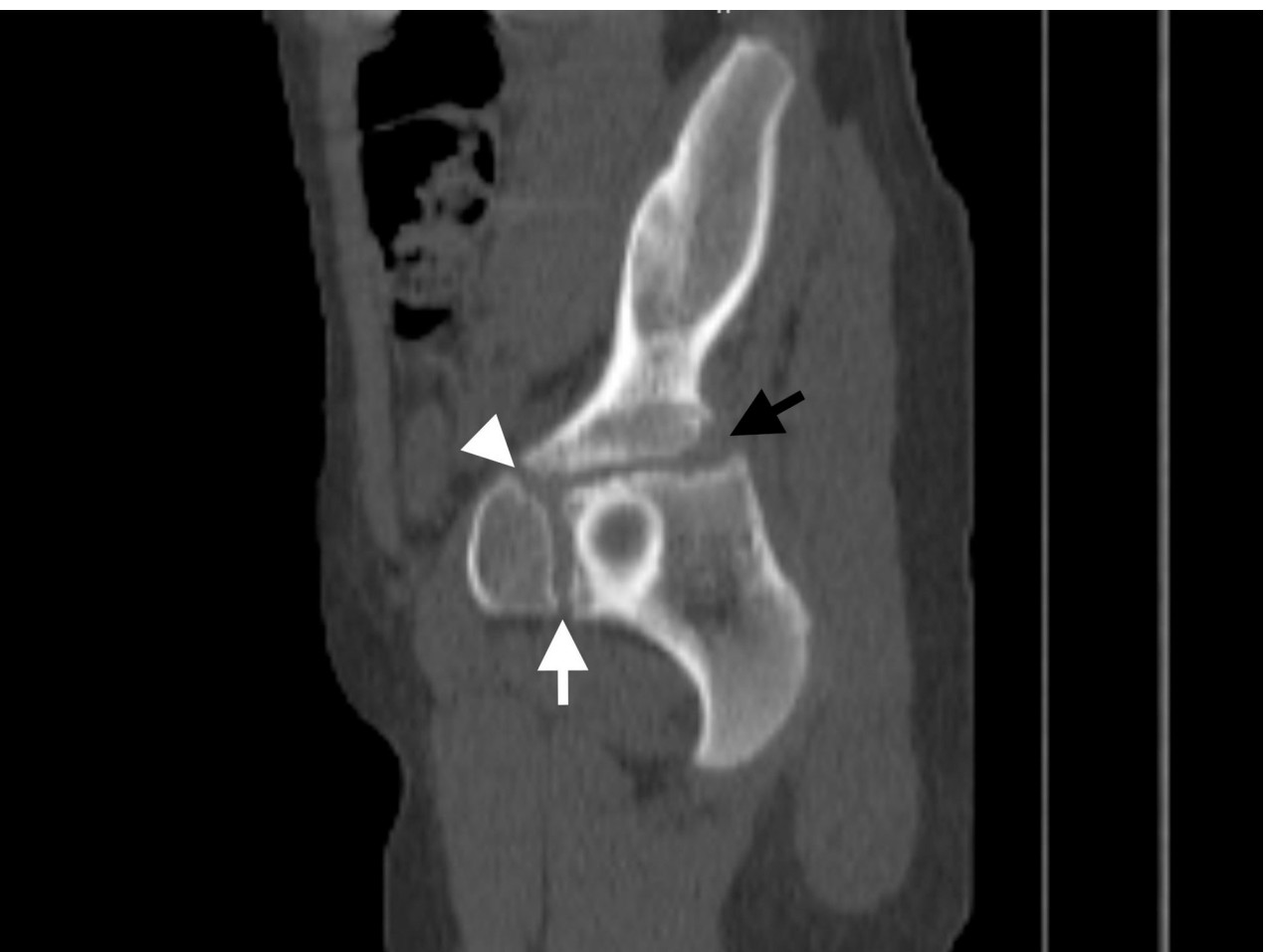

**Fig 6. For a patient who is an 8-year-old girl, the sagittal plane reformatted computed tomography image shows that all three (white arrow: Ischiopubic, black arrow ilioischial, white arrowhead: Iliopubic) synchondrosis regions are open.**

These studies reported that spheno-occipital synchondrosis is closed at the age of 18 and above and is open under the age of 10 [20–22]. Again, similarly, there are studies in which age estimation can be made with medial clavicular epiphysis. One study reported that the clavicular epiphysis began to be observed between the ages of 11–21, that there was a partial closure between the ages of 16–21, and that full closure began to appear around the ages of 18–19. In another study, it was reported that there was nonunion in the age range of 8–17, there was partial union at the ages of 15–20, and there was complete union at the ages of 20 [23,24]. In a study on the fusion of the hyoid bone, an increase in the frequency of hyoid bone fusion was observed after 20 years of age [25]. In addition, Norouzi et al. reported that age determination could be performed between 12 and 20 years of age by fusion of the iliac crest [26]. It has been reported that scapular fusion can only be a parameter that can be used for age determination around the age of 18. Computed tomography studies can be considered a useful method for age determination. Therefore, it may provide additional information to the relevant clinicians for age determination [27]. Our study showed that forensic radiological skeletal age estimation could be performed using pelvis computed tomography using ischiopubic-ilioischial-iliopubic synchondrosis (Tables 1 and 2, and Figs 7 and 8).

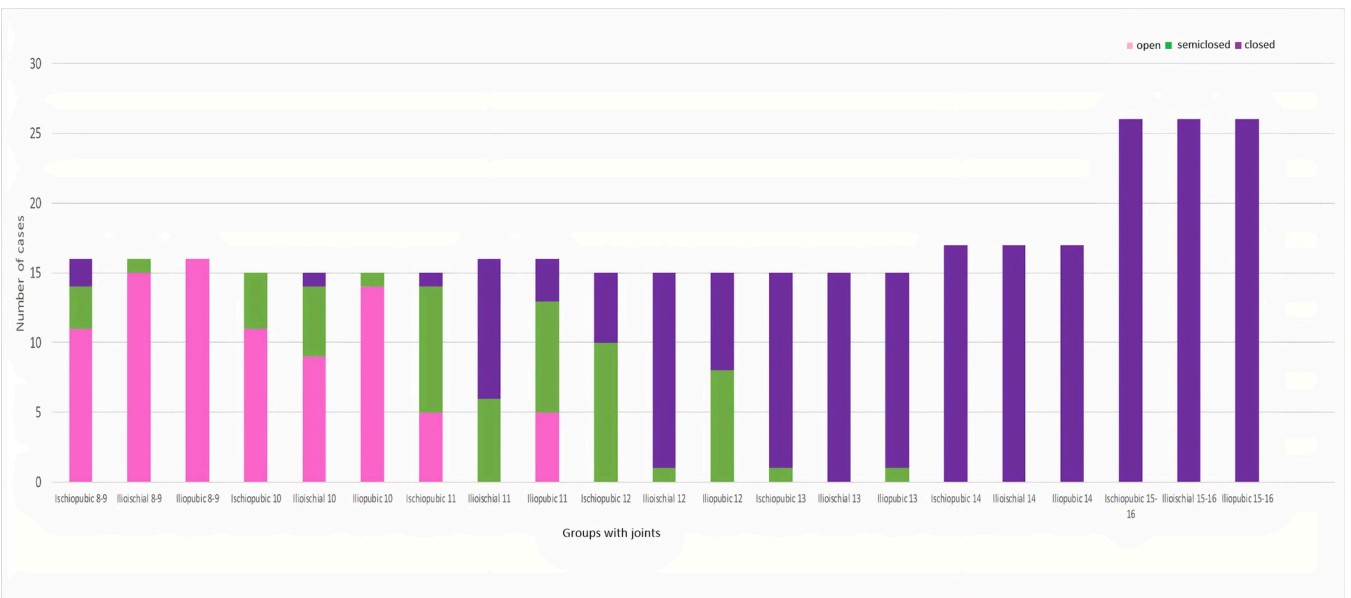

**Fig 7. Frequency distributions by the group for ischiopubic, ilioischial, and iliopubic synchondrosis regions in girls.** 0: Pink (open), 1: Green (semiclosed), and 2: Purple (closed).

The general approach to age estimation is made according to the degree of closure of the bones' epiphyseal lines. The degree of bone development is expressed as bone age, and evaluation is performed by comparison with standard cases. In a case with normal bone maturation, bone age is equal to chronological age. Os coxa develops in three primary ossification centers that ossify from a hyaline cartilage template in the 2nd to the sixth month of embryological development. Ossification begins sequentially with the superior ilium, followed by the posterior ischium, and ends with the anterior pubis. In a study by Grissom et al., the mean age of the closure of ischiopubic ramus synchondrosis was reported to be 12. In a recent study by Gregory et al. on CT images, ischiopubic ramus synchondrosis closure was very important in

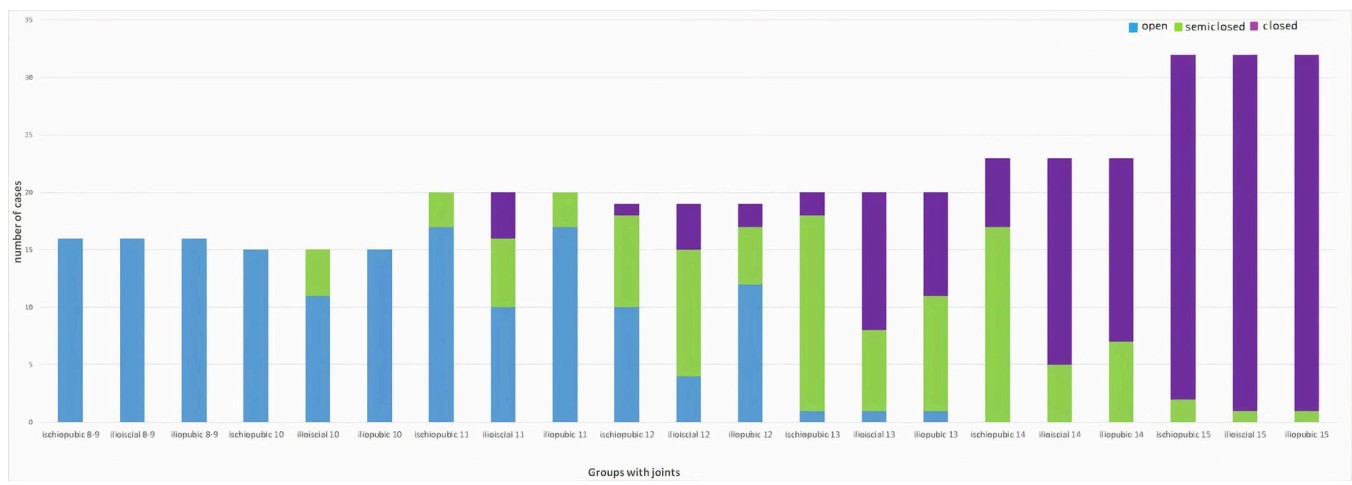

**Fig 8. In boys, there is frequency distributions by groups for ischiopubic, ilioischial, and iliopubic synchondrosis.** 0: Blue (open), 1: Green (semiclosed), and 2: Purple (closed).

age determination. It has been shown that ischiopubic ramus synchondrosis begins and continues at age 7–13 in boys and 4–9 years in girls [28,29].

It was previously reported that triradiate ossification occurs at approximately the age of 15 and over [19]. Closure of the triradiate complex is earlier in women than in men [30,31]. The tri-radiate complex consists of ischiopubic-ilioischial-iliopubic synchondrosis. In our study, parts of the triradiate complex (ischiopubic-ilioischial-iliopubic synchondrosis) were evaluated separately, and our results were found to be compatible with the literature. In a study with Korean children, ossification of the TRC was completed earlier in girls than in boys [32,33]. In our study, similar to a study conducted in Korea, it was observed that ischiopubic-ilioischial-iliopubic synchondrosis was closed earlier in girls than in boys (Figs 7 and 8). It has been stated that the fusion of the tri-radiate complex is completed at the age of 16–17 years [33]. Our study observed that the fusion of the tri-radiate complex was completed at the age of 15 (Tables 1 and 2). It is known that radiological bone age may differ according to ethnic origins [34]. For this reason, it is crucial to obtain results specific to societies by researching each society.

The chronological age has become increasingly important for children. Because many rights and responsibilities are associated with legal age limits and the adoption of the Convention on the Rights of the Child and its translation into national legislation, for example, governments may allow children to marry, consent to sexual relations, and to accept or refuse health services before reaching adulthood and remove age restrictions on military service or recruitment. The age at which criminal responsibility is acquired can precede the legal age-based definition of adulthood and range from 7 to 16 years [10].

According to the laws of Turkey, individuals under 12 years of age do not have criminal responsibility, and it is stated that a decision should be made by examining whether there is a criminal liability in the age range of 12–15 years. Again, within the scope of the criminal law, whether the courts can defend themselves in terms of body and soul according to the crime perpetrated by considering the individuals' age is evaluated. Additionally, in some cases, such as working with civil law, marriage, inheritance, and the capacity to act, the courts may grant some rights for individuals over the age of 15 [35,36]. In almost every country of the world, laws are similar to those of our country and are enforced. Our study may be useful since it covers the 8-16-year-old age groups and can be used to conduct age estimation (Figs 7 and 8).

One advantage of age determination methods using the pelvis is that this part of the body is very resistant to decay. The acetabulum is generally observed as preserved in the found skeletons [37,38]. A study on the damage done by the jackals and dogs on corpses between 4 hours and 52 months showed the following. Stage 0: There is no evidence of disarticulation and soft tissue loss in this early stage (postmortem four hours-14 days). Stage 1: There is a fractured ventral thorax characterized by loss of the sternum, destruction of sternum ribs, evisceration, loss of the scapula, partial/total clavicle loss, and loss of one or both upper extremities (characteristic after 22 days-2.5 months). Stage 2: In addition to Stage 1, it includes the lower extremities (postmortem 2–4,5 months). Stage 3: Except for the spine, all skeleton parts are broken, severely injured, and destroyed. At this stage, the bones are scattered over a distance ranging from 3 to 91 m (postmortem 2–11 months). Stage 4: Total disarticulation (postmortem 6–52 months) [39]. As evidence shows, pelvis bones can withstand postmortem period effects and animal attacks for a more extended period. We believe that age estimation can be made on corpses with no extremities, a damaged chest, or only pelvic bones that are accessible (Figs 1–6).

In order to determine the age of unidentified persons, bone morphologies of dead soldiers were investigated, and it was found that the formation of lipping and exostoses around the ischium appeared to be a gradual and erratic process and that such changes tended to become pronounced in the latter part of the fourth decade. However, it should be apparent that they

are not accurate age indicators [40]. In our study, ischiopubic-ilioischial-iliopubic synchondrosis in age-determined children under the age of 17 was evaluated in both sexes. The morphology of ischiopubic-ilioischial-iliopubic synchondrosis may be utilized for the forensic determination of the bone age of unidentified people under the age of 17.

In many countries, increased immigration has led to an increase in the number of foreigners who are unable to provide documents on birth dates. As a result of this development, age estimation using forensic practices has become an essential part of forensic practices [41]. Various restrictions were encountered in our study. First, our cases are composed of groups of a single ethnic origin. Second, pelvis computed tomography images taken in our study were used. We consider that the same images can be obtained by magnetic resonance imaging. Even though the process takes a long time with magnetic resonance imaging, we believe that this method can be utilized for children not to be exposed to radiation.

## Conclusion

In our study, the ischiopubic-ilioischial-iliopubic joints were open in those aged nine and under and closed in those aged 15 and above. Ilioischial, ischiopubic, and iliopubic synchondrosis were observed to close earlier in girls than in boys. As a result of our study, it was concluded that by examining the ischiopubic-ilioischial-iliopubic synchondrosis, information could be obtained in terms of forensic radiological bone age. Our study will be beneficial in the 8-16-year-old age group if used with the other forensic radiological skeletal age estimation methods. In addition, our study can be used to determine the radiological bone age in cases with wrist bone abnormalities or wrist amputation.

## Supporting information

**S1 File.**
(XLSX)

## Author Contributions

**Data curation:** Burak Gümüş, Erdal Karavaş, Onur Taydaş.

**Formal analysis:** Burak Gümüş, Onur Taydaş.

**Funding acquisition:** Burak Gümüş.

**Investigation:** Burak Gümüş, Erdal Karavaş.

**Methodology:** Burak Gümüş, Erdal Karavaş.

**Project administration:** Burak Gümüş.

**Resources:** Burak Gümüş, Erdal Karavaş.

**Software:** Burak Gümüş, Erdal Karavaş, Onur Taydaş.

**Supervision:** Onur Taydaş.

**Visualization:** Burak Gümüş, Erdal Karavaş, Onur Taydaş.

**Writing – original draft:** Burak Gümüş.

**Writing – review & editing:** Burak Gümüş.

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
