## [Decision Letter · Decision Letter 0]

16 Sep 2021

PONE-D-21-24189Can Forensic Radiological Skeletal Age Estimation be Performed by Examining Ischiopubic-ilioischial-iliopubic synchondrosis in Computed Tomography Images?PLOS ONE

Dear Dr. Gümüş,

Thank you for submitting your manuscript to PLOS ONE. After careful consideration, we feel that it has merit but does not fully meet PLOS ONE’s publication criteria as it currently stands. Therefore, we invite you to submit a revised version of the manuscript that addresses the points raised during the review process.

We look forward to receiving your revised manuscript.

Kind regards,

Alok Atreya

Academic Editor

PLOS ONE

Journal Requirements:

2. Please confirm all data were analyzed anonymously and that the IRB waived any requirements for patient or parental consent.

Reviewers' comments:

Reviewer's Responses to Questions

**Comments to the Author**

1. Is the manuscript technically sound, and do the data support the conclusions?

Reviewer #1: Partly

Reviewer #2: Yes

Reviewer #3: Yes

2. Has the statistical analysis been performed appropriately and rigorously? 

Reviewer #1: Yes

Reviewer #2: Yes

Reviewer #3: Yes

3. Have the authors made all data underlying the findings in their manuscript fully available?

Reviewer #1: No

Reviewer #2: No

Reviewer #3: Yes

4. Is the manuscript presented in an intelligible fashion and written in standard English?

Reviewer #1: No

Reviewer #2: Yes

Reviewer #3: Yes

5. Review Comments to the Author

Reviewer #1: The paper is in a very raw stage, it needs extensive revision.

There are no page numbers in the manuscript? Please take upmost care in preparing the manuscript and please double check every part of the manuscript.

1. In the abstract, why the 'Instrument and methods' was used? Is there any instrument?

Please give some more pictures as examples of the CT images showing various stages of fusion of the synchondrosis. Please explain more methods in the abstract.

"Two hundred sixty-three children (118 girls and 145 boys) between the ages of 8 and 16 without any health problems participated in the study"- this is a part of methodology and not the results. Please change '8 and 16' to '8 and 16 years'

Why are you mentioning "Discussion and results', why not 'Results and Discussion' . Please use standard format given by the journal.

2.The introduction is very superficial, please write detailed introduction. Please use recent relevant studies and expand the introduction.

"Age is one of the physical features, along with gender,.... Age is not a feature? and please replace gender with 'sex'. Please use appropriate words throughout as the inappropriate words may change the meaning of the sentences.

3. In the methodology, 'According to the defined UNICEF age limits...',you have made different age specific groups, but the sample size guidelines were not followed, statistically, these are not valid without sizable sample.

Please mention dates/years of the examination of the patients.

4. Please discuss the sex differences as obtained in the study in light with other studies.

5. The conclusion should contain concrete findings of the study according to the title as well the objective of the study. Also mention, What is the percentage of accuracy of the determination of age from these dimensions for the mentioned ages?

Why are you mentioning 'results' in stead of 'conclusion'?

6. In the reference section, The format of the references is not uniform, please use journal's guidelines for the same. Please do not use papers from predatory journals such as reference no. 6, 13 etc. Use only which are indexed in PUBMED, SCOPUS and SCI journals.

Reviewer #2: Thank you for presenting this significant study for forensic age estimation.

Please find below some suggestions to help improve the article.

The language needs to be evaluated in further detail. Please use consistent tense forms.

The results needs to be further elaborated. What were the significance values for the Chi-square tests. The p-values should be correlated along with the significance values.

Discussion should focus on comparison with other studies and their findings and discussing the relevance of similarities and differences in these studies.

Most of the presently submitted discussion section, including validating Age estimation as well as CT studies, legal provisions, ability of pelvis to withstand trauma, etc should all be detailed in the Introduction and referenced in brief in discussion, if required.

Please make the suggested changes and resubmit.

Reviewer #3: The manuscript is technically sound with appropriate statistical analysis. Manuscript is in good standard English. Data underlying the findings in the manuscript is fully available. Please do strict in research ethics.

6. PLOS authors have the option to publish the peer review history of their article (what does this mean?). If published, this will include your full peer review and any attached files.

Reviewer #1: No

Reviewer #2: **Yes: **Rijen Shrestha

Reviewer #3: No

---

## [Author Response · Author response to Decision Letter 0]

14 Dec 2021

Reviewer 1:

- In the abstract, why the 'Instrument and methods' was used? Is there any instrument?

• The "instrument" phrase has been removed and corrected as "Methods."

- Please give some more pictures as examples of the CT images showing various stages of fusion of the synchondrosis.

• New images are added.

- Please explain more methods in the abstract.

• Added more descriptive information to the Methods section.

- "Two hundred sixty-three children (118 girls and 145 boys) between the ages of 8 and 16 without any health problems participated in the study"- this is a part of methodology and not the results. 

• This part is transferred to the methodology section. 

- Why are you mentioning "Discussion and results', why not 'Results and Discussion'. Please use standard format given by the journal.

• The "Result" statement was removed from the abstract section and the "Conclusion" statement was added instead.

- Please change '8 and 16' to '8 and 16 years'

• ok

- "Age is one of the physical features, along with gender,.... Age is not a feature? 

• This part is removed.

- Please replace gender with 'sex'. Please use appropriate words throughout as the inappropriate words may change the meaning of the sentences.

• The word "gender" has been replaced with the word "sex."

- "According to the defined UNICEF age limits," 

• This part is removed.

- Please mention dates/years of the examination of the patients.

• ok

- Please discuss the sex differences as obtained in the study in light with other studies.

• New additions were made to the discussion.

- Why are you mentioning 'results' in stead of 'conclusion'?

• The "Result" statement was removed from the abstract section and the "Conclusion" statement was added instead. 

- In the reference section, The format of the references is not uniform, please use journal's guidelines for the same. Please do not use papers from predatory journals such as reference no. 6, 13 etc. 

• These references were removed and new ones were added in their place.

- The introduction is very superficial, please write detailed introduction.

• Additions were made by detailing the introduction section.

- The conclusion should contain concrete findings of the study according to the title as well the objective of the study. Also mention, What is the percentage of accuracy of the determination of age from these dimensions for the mentioned ages?

• The conclusion section was arranged to be more consistent with the findings.

- In the reference section, The format of the references is not uniform, please use journal's guidelines for the same.

• References were reviewed according to journal guidelines.

Reviewer 2

Reviewer #2: Thank you for presenting this significant study for forensic age estimation.

Please find below some suggestions to help improve the article.

- The language needs to be evaluated in further detail. Please use consistent tense forms.

- The results needs to be further elaborated. What were the significance values for the Chi-square tests. The p-values should be correlated along with the significance values.

OK

- Discussion should focus on comparison with other studies and their findings and discussing the relevance of similarities and differences in these studies.

With new references, the discussion has been expanded upon your suggestion.

- Most of the presently submitted discussion section, including validating Age estimation as well as CT studies, legal provisions, ability of pelvis to withstand trauma, etc should all be detailed in the Introduction and referenced in brief in discussion, if required.

With the new resources, the introduction part has been expanded according to your suggestions.

Reviewer 3

reviewer #3: The manuscript is technically sound with appropriate statistical analysis. Manuscript is in good standard English. Data underlying the findings in the manuscript is fully available. Please do strict in research ethics.

Ethics committee approval document is attached.

---

## [Decision Letter · Decision Letter 1]

25 Mar 2022

Can Forensic Radiological Skeletal Age Estimation be Performed by Examining Ischiopubic-ilioischial-iliopubic synchondrosis in Computed Tomography Images?

PONE-D-21-24189R1

Dear Dr. Gümüş,

We’re pleased to inform you that your manuscript has been judged scientifically suitable for publication and will be formally accepted for publication once it meets all outstanding technical requirements.

Kind regards,

Qingzhong Liu, PhD

Academic Editor

PLOS ONE

Additional Editor Comments (optional):

Reviewers' comments:

Reviewer's Responses to Questions

**Comments to the Author**

1. If the authors have adequately addressed your comments raised in a previous round of review and you feel that this manuscript is now acceptable for publication, you may indicate that here to bypass the “Comments to the Author” section, enter your conflict of interest statement in the “Confidential to Editor” section, and submit your "Accept" recommendation.

Reviewer #1: All comments have been addressed

2. Is the manuscript technically sound, and do the data support the conclusions?

Reviewer #1: Yes

3. Has the statistical analysis been performed appropriately and rigorously? 

Reviewer #1: Yes

4. Have the authors made all data underlying the findings in their manuscript fully available?

Reviewer #1: Yes

5. Is the manuscript presented in an intelligible fashion and written in standard English?

Reviewer #1: Yes

6. Review Comments to the Author

Reviewer #1: (No Response)

7. PLOS authors have the option to publish the peer review history of their article (what does this mean?). If published, this will include your full peer review and any attached files.

Reviewer #1: No

---

## [Editor Report · Acceptance letter]

30 Mar 2022

PONE-D-21-24189R1 

Can Forensic Radiological Skeletal Age Estimation be Performed by Examining Ischiopubic-ilioischial-iliopubic synchondrosis in Computed Tomography Images? 

Dear Dr. Gümüş:

I'm pleased to inform you that your manuscript has been deemed suitable for publication in PLOS ONE. Congratulations! Your manuscript is now with our production department. 

Kind regards, 

on behalf of

Dr. Qingzhong Liu 

Academic Editor

PLOS ONE